# The Adhesion GPCR VLGR1/ADGRV1 Regulates the Ca^2+^ Homeostasis at Mitochondria-Associated ER Membranes

**DOI:** 10.3390/cells11182790

**Published:** 2022-09-07

**Authors:** Jacek Krzysko, Filip Maciag, Anna Mertens, Baran Enes Güler, Joshua Linnert, Karsten Boldt, Marius Ueffing, Kerstin Nagel-Wolfrum, Martin Heine, Uwe Wolfrum

**Affiliations:** 1Institute of Molecular Physiology (imP), Molecular Cell Biology, Johannes Gutenberg University Mainz, 55128 Mainz, Germany; 2Institute for Developmental Biology and Neurobiology (IDN), Functional Neurobiology, Johannes Gutenberg University Mainz, 55128 Mainz, Germany; 3Institute for Ophthalmic Research, University of Tuebingen, 72076 Tuebingen, Germany

**Keywords:** adhesion GPCR, mitochondria-associated ER membranes (MAM), mitochondria-endoplasmic reticulum contact sites (MERCS), Ca^2+^ transient at ER and mitochondria, Ca^2+^ homeostasis

## Abstract

The very large G protein-coupled receptor (VLGR1, ADGRV1) is the largest member of the adhesion GPCR family. Mutations in VLGR1 have been associated with the human Usher syndrome (USH), the most common form of inherited deaf-blindness as well as childhood absence epilepsy. VLGR1 was previously found as membrane–membrane adhesion complexes and focal adhesions. Affinity proteomics revealed that in the interactome of VLGR1, molecules are enriched that are associated with both the ER and mitochondria, as well as mitochondria-associated ER membranes (MAMs), a compartment at the contact sites of both organelles. We confirmed the interaction of VLGR1 with key proteins of MAMs by pull-down assays in vitro complemented by in situ proximity ligation assays in cells. Immunocytochemistry by light and electron microscopy demonstrated the localization of VLGR1 in MAMs. The absence of VLGR1 in tissues and cells derived from VLGR1-deficient mouse models resulted in alterations in the MAM architecture and in the dysregulation of the Ca^2+^ transient from ER to mitochondria. Our data demonstrate the molecular and functional interaction of VLGR1 with components in MAMs and point to an essential role of VLGR1 in the regulation of Ca^2+^ homeostasis, one of the key functions of MAMs.

## 1. Introduction

VLGR1 (very large G protein-coupled receptor-1), also known as ADGRV1, GPR98, and MASS, has a molecular weight of up to 700 kDa, making it the largest member of the 33 adhesive G protein-coupled receptors (ADGRs), a unique subfamily of the GPCR superfamily [1]. The molecular architecture of ADGRs is characterized by an extended large extracellular domain (ECD), a seven-span transmembrane domain (7TM) and a short intracellular domain (ICD) (Figure 1A). Autoproteolysis at the highly conserved GPCR proteolytic site (GPS) in the GAIN (autoproteolysis-inducing) domain positioned next to the 7TM results in a bipartite receptor molecule [2]. The resulting molecules consist of the N-terminal fragment (NTF), which contains characteristic cell-adhesive protein motifs and the G protein signaling 7TM-containing C-terminal fragment (CTF).

VLGR1 is expressed almost ubiquitously in humans and mice with high expression levels in the nervous system [4]. In humans, mutations of the VLGR1 gene cause Usher syndrome (USH), the most common form of hereditary deaf-blindness [5]. Haploinsufficiency in VLGR1/ADGRV1 has been associated with childhood absence epilepsy [4,6,7]. In mice, defects in Vlgr1 lead to audiogenic seizures [4,6,7].

We have previously shown that VLGR1 is a part of membrane–membrane adhesion complexes associated with the photoreceptor primary cilium and the ankle-links of stereocilia of the developing hair cells [8]. In addition, VLGR1 functions as a metabotropic mechanoreceptor in focal adhesions [9]. More recently, our affinity proteomics-based studies provided the first evidence that VLGR1 associates with molecules of intracellular membrane networks, namely of the endoplasmic reticulum (ER) and the mitochondria-associated ER membranes (MAMs) [3,8].

MAMs, also named MERCs (mitochondria-ER contacts), are specialized subcellular compartments that are shaped by specific subdomains of the ER surface juxtaposed to the outer mitochondria membrane. MAMs are composed of a characteristic set of molecules which guarantees the juxtaposition between these organelles and determines several intracellular processes, such as Ca^2+^ and lipid homeostasis, immune response or autophagy [10]. The MAM contact sites are dynamic and the dysfunction of MAMs has been associated with various neurodegenerative disorders, such as Alzheimer’s or Parkinson’s disease [11,12].

Here, we demonstrate the localization and functions of an aGPCR in the ER, particularly in the MAM complex. We show that VLGR1 contributes to the arrangement of a protein complex that is crucial for regulating Ca^2+^ flux from the ER into the MAM interface and the uptake through the outer mitochondria membrane. We demonstrate that the arrangement of the MAMs is altered in Vlgr1-deficient neurons. Further, live cell imaging of Ca^2+^ shows that the Ca^2+^ fluxes from the ER and into mitochondria are severely impaired in primary astrocytes from Vlgr1-deficient *Vlgr1*del7TM. The morphological changes and disrupted Ca^2+^ homeostasis observed in the absence of VLGR1 from the MAM possibly contribute to the pathophysiology of the disorders caused by defects in VLGR1 in USH2 and childhood absence epilepsy patients.

## 2. Materials and Methods

### 2.1. Animals

All experiments were performed in compliance with guidelines established by Association for Research in Vision and Ophthalmology. Mice were kept under 12/12 h light/dark cycles, food and water ad libitum. *Vlgr1*del7TM mice harbor a nonsense mutation in Vlgr1, V2250X, which results in loss of Vlgr1’s transmembrane and cytoplasmic domain [13]. Breeding background of *Vlgr1*del7TM mice was the C57BL/6 strain. The use of mice in research was approved by District administration Mainz-Bingen, 41a/177-5865-§11 ZVTE, 30.04.2014. Zebrafish were obtained from the Institute of Molecular Biology (IMB). Porcine eyes were obtained from the local slaughterhouse.

### 2.2. Antibodies

The following primary antibodies were used: rabbit anti-Nogo-A/RTN4 (AHP1799, Biorad, Feldkirchen, Germany), mouse anti-TOM20 (Santa Cruz, CA, USA, sc-17764), mouse anti-SIGMA1R (Santa Cruz, sc-137075), rat anti-SIGMA1R (Merck, Darmstadt, Germany, ICR-SIG1R-A), mouse anti-ACSL4 (Santa Cruz, sc-365230), mouse anti-Myc (Cell Signaling, Danvers, MA, USA, 2276), rabbit anti-HA (Sigma-Aldrich, St. Louis, MO, USA, H6908), rat anti-HA (Roche, Basel, Switzerland, 3F10), rabbit anti-VLGR1, raised against the against the C-terminus of murine VLGR1 (amino acids 6198–6307) and previously characterized in [9,14], rabbit anti-α-Tubulin (Abcam, Cambridge, UK, DM1A), rabbit anit-COX IV (NEB, Ipswich, MA, USA, 3E11), rabbit anti-GFAP (Dako Agilent, Santa Clara, CA, USA, Z0334), rat anti-RFP (Chromotek, Planegg, Germany, 5F8), mouse anti-CLIMP63, (Enzo Life Sciences, G1/296), goat anti-Pericentrin 2 (Santa Cruz, C-16), mouse anti-Arl13b (Abcam, Cambridge, UK, ab136648), goat anti-Centrin 2 [15]. Secondary antibodies conjugated Alexa 488, Alexa 555, Alexa 568 or Alexa 647 were purchased from Molecular Probes (Life Technologies, Darmstadt, Germany) or Rockland Inc. (Gilbertsville, PA, USA). Nuclear DNA was stained with DAPI (4′,6-diamidino-2-phenylindole, 1 mg/mL: Sigma-Aldrich).

### 2.3. Cell Lines

We used the following cell lines: HEK293T cells constitutively expressing Simian virus 40 (SV40) large T antigen isolated from human embryonic kidney tissue were mainly used as highly transfectable cells in assays, such as tandem affinity purifications (TAPs, Section 2.6). hTERT-RPE-1 are human telomerase reverse transcriptase (hTERT)-immortalizeretinal cells from the retinal pigment epithelium (RPE) of the human eye [8]. The genetically stable, nearly diploid hTERT-RPE1 cells have been previously used in studies on VLGR1 and were used here in additional complementary TAPs [9]. HeLa cells, epithelial cells isolated from cervical carcinoma were used for the analysis of the ER-mitochondria interface. All cell lines were initially purchased from American Type Culture Collection (ATCC).

### 2.4. Cell Culture

HEK293T and HeLa were cultured in Dulbecco’s modified Eagle’s medium (DMEM), and hTERT-RPE1 cells were cultured in DMEM-F12, respectively, all supplemented with 10% fetal bovine serum (FBS) (ThermoFisher Scientific, Waltham, MA, USA). Cells were transfected with GeneJuice^®^ (Merck Millipore, Darmstadt, Germany) according to manufacturer’s instructions. Primary astrocyte cultures were prepared from cerebral cortices of C57BL/6 WT, Vlgr1/del7TM mice, as previously described [16]. In brief, mouse pups from C57BL/6 WT, Vlgr1/del7TM were dissected postnatal day 0 (PN0), and cerebral cortices were collected in 1x HBSS (ThermoFisher Scientific, Waltham, MA, USA) medium contains Dnase/trypsin (Merck, Darmstadt, Germany) for enzymatic dissociation. Additionally collected cortices were mechanically dissociated by 10 mL and 5 mL pipettes, respectively. Single cell suspensions were cultured in DMEM/10% FBS/2% penicillin/streptomycin (ThermoFisher Scientific, Waltham, MA, USA) and growth medium was changed on day 1, day 2, and day 7. Upon confluency, oligodendrocytes and neurons were removed by shaking the plates. To remove microglia cells, trypsin and DNase were added to dishes and cells were passed over successive bacterial grade dishes. Primary astrocytes were cultured for 14 days in complete growth medium and used for downstream experiments.

### 2.5. DNA Constructs

VLGR1_CTF (Uniprot ID Q8WXG9-1, aa 5891-6306) sequence was used for VLGR1 constructs. For tandem affinity purifications, Strep II-FLAG (SF)-tagged human VLGR1_CTF was used. The SF-tag was N-terminally and C-terminally fused in VLGR1_CTF. For HA-and RFP-tagged constructs, VLGR1_CTF was subcloned into a HA/DEST vector (pDEST520; Invitrogen, Waltham, MA, USA) and an RFP/DEST vector (pDEST733; Invitrogen). The SIGMA1R-Myc (S1R-Myc) construct was a kind gift from Christian Behl (University Medical Center Mainz). Indicators for calcium imaging were pCMV CEPIA2mt and pCMV G-CEPIA1er [17].

### 2.6. Tandem Affinity Purification (TAP) and Mass Spectrometry

The tandem affinity purification (TAP) and mass spectrometry (LC-MS/MS) analysis were performed as previously described [8,18]. The two Strep II/FLAG-(SF)-tagged VLGR1_CTF (Figure 1B) were expressed in HEK293T cells or hTERT-RPE1 cells, respectively for 48 h, lysed and cleared by centrifugation. Mock-treated cells were used as controls. Subsequently supernatants were subjected to a two-step purification on Strep-Tactin^®^ Superflow^®^ beads (IBA, Göttingen, Germany) and anti-FLAG M2 agarose beads (Merck). In these steps competitive elutions were achieved by desbiothin (IBA) and FLAG^®^ peptide (Merck), respectively. Methanol-chloroform precipitated eluates were subjected to liquid chromatography coupled with tandem mass spectrometry (LC-MS/MS).

### 2.7. Data Processing

Obtained raw MS spectra were searched against the human SwissProt database using Mascot and results were verified by Scaffold (version Scaffold 4.02.01, Proteome Software Inc., Portland, OR, USA) for the validation of MS/MS-based peptide and polopeptide identifications. Mass spectrometry results for the tow VLGR1_CTF fragment were compared to the according data for mock-transfected cells and to common control TAPs of the RAF1-protein [18]. Proteins in mock and RAF1 datasets were excluded for subsequent analysis. VLGR1_CFT preys were used as input for the Cytoscape plugins STRING, ClueGO and the STRAP software according to their gene names based on HGNC. Confidence (score) cutout 0.40 and maximum number of interactors 0 were set as parameters for STRING analysis. Gene Ontology (GO) term enrichment analysis was performed by ClueGO v2.3.3. Network specificity was set to default (medium).

### 2.8. RFP-Trap^®^/Myc-Trap^®^

Nanobody RFP-Trap^®^/Myc-Trap^®^ agarose beads (ChromoTek) were used for precipitation assays according to the manufacturer’s protocol. Briefly, RFP/Myc-tagged proteins were expressed in HEK293T cells (24 h). For cell lysis, Triton X-100 lysis buffer (50 mM Tris–HCl pH 7.5, 150 mM NaCl, and 0.5% Triton X-100) containing protease inhibitor cocktail (PI mix; Roche) was used. 10% of total cell lysates were separated for input. Remaining lysates were incubated on equilibrated beads for 2 h at 4 °C under constant rotation. After washing the beads with dilution buffer (10 mM Tris-HCl pH 7.5, 150 mM NaCl, 0.5 mM EDTA, precipitated proteins were eluted with SDS buffer. Samples were analyzed by SDS-PAGE and western blotting.

### 2.9. Immunocytochemistry and Fluorescence Microscopy

Cells were fixed with 4% paraformaldehyde in PBS for 10 min at RT, washed with PBS, permeabilized with PBST (0.1% Triton-X100 (Roth, Karlsruhe, Germany)) 5 min at RT and blocked with 0.1% ovalbumin, 0.5% fish gelatine in PBS for 45 min at RT. Primary antibodies were incubated overnight at 4 °C, followed by washing with PBS and secondary antibody incubations 1 h at RT. After a final washing cycle (3 × 10 min), cells were mounted with Mowiol 4.88 (Hoechst) and analyzed with a Leica DM6000B microscope (Leica, Wetzlar, Germany). Images were acquired by sCMOS K5 camera (Leica-microsystems GmbH) with a Leica HCX PL 63x/1.32 objective and processed using the Leica Application Suit X, Adobe Photoshop CS^®^ (Adobe Systems, San Jose, CA, USA), and Image J.

### 2.10. Isolation of Mitochondria-Associated Membrane Fraction

MAMs were isolated from HEK293T cells transfected with VLGR1_CTF_HA according to the protocol by Wieckowski et al. [19]. The workflow of the isolation of MAMs are illustrated in the Appendix A. In brief, 48 h after transfection cells were harvested and homogenized in IBcells1 buffer (225 mM D-Mannitol, 75 mM sucrose and 30 mM-Tris-HCl and 0.1 mM EGTA, pH 7.4) using a Dounce homogenizer (B. Braun Melsungen, Melsungen, Germany). To remove unbroken cells, plasma membranes and nuclei a centrifugation step applied at 700× *g* for 5 min using an Eppendorf Centrifuge 5430 R. Supernatants were carefully collected and additional centrifugation step at 7000× *g* for 10 min were carried out to obtain the supernatants for cytosolic fractions containing ER, lysosomes, and microsomes (Mic), and the pellet for crude mitochondria fraction. In a 1st step, supernatants were transferred to new centrifugation tubes and centrifugated at 20,000× *g* for 30 min in an Eppendorf Centfigure 5910 R (Eppendorf FA-6 × 50 rotor) to separate the ER and microsomes. Pure ER pellets were obtained with further ultra-centrifugation at 100,000× *g* for 1 h (Beckman Coulter OptimaTLX, SW 40 Ti Swinging-Bucket Rotor, Brea, CA, USA) and resuspended in MRB buffer (250 mM D-Mannitol, 25 mM HEPES, 1 mM EGTA). In a 2nd step, pellets for crude mitochondria fractionation were resuspended in IBcells-2 buffer (225 mM D-Mannitol, 75 mM sucrose, 30 mM Tris-HCl) and centrifugated at 7000× *g* for 5 min in two rounds to remove contaminants. The obtained pellets of crude the mitochondrial fraction were then resuspended in MRB buffer and added on 8 mL cushion of 30% Percoll in 225 mM D-Mannitol, 25 mM HEPES, 1 mM EGTA for the subfractionation of MAMs and pure mitochondria. After ultra-centrifugation at 95,000× *g* for 30 min, MAM and mitochondria fractions could be distinguished as two separated bands in the tubes. Mitochondria fractions were collected from the band localizing close the bottom of the tube and MAM fractions were collected from the band above. Collected mitochondria fractions were resuspended in buffer and centrifugated for 10 min at 7000× *g* to remove potential MAM contaminants. Supernatants were discarded and pellets resuspended in MRB buffer as the pure mitochondria. Finally, MAM fractions from previous steps were centrifugated at 100,000× *g* for 30 min to remove mitochondria contaminants. Supernatants were removed and pellets resuspended in MRB buffer as pure MAM fractions. All obtained fractions were analyzed by Western blot applying markers of the different subcellular fractions.

### 2.11. Co-Localization Analysis

Pearson correlation coefficient (R) was used to determine the degree of colocalization between VLGR1_CTF_RFP and ER-mitochondrial interface in cells [20]. The correlation value has a range from +1 to −1. A value of 0 indicates no association, greater than 0 indicates a positive association and less than 0 indicates a negative association between the two variables. The Pearson coefficient was calculated using the Coloc 2 plugin of ImageJ (https://imagej.nih.gov/ij/ (accessed on 19 December 2019)). To determine the localization of the interface of mitochondria and ER the immunofluorescences staining for TOM20 and Nogo-A/RTN4 [21] were subtracted by applying “subtract function” in Adobe Photoshop^®^ CS5 V5 (AP) (Adobe Systems). For this, immunofluorescence signals for the ER marker Noga-A and the mitochondria marker TOM20 in the whole image were screened via AP function: screen and subtracted via AP function: subtract, leaving only the signals with an overlap (Noga-A/TOM20 co-staining), which allowed us to demonstrate the Noga-A and TOM20 signal interface of ER and mitochondria, roughly representing the MAM compartment. We additionally validated the co-localization of immunostained molecules by fluorescence intensity line plots. For this, lines were drawn across to the region of interest (ROI) and pixel intensities were analyzed using Leica Application Suit X (Leica-microsystems). The co-localization of VLGR1_CTF_RFP with ER-mitochondrial markers were determined by the peak values of the signals in fluorescence intensity plots.

### 2.12. Transmission Electron Microscopy (TEM)

For conventional TEM dissected eyeballs and brains were pre-fixated for 2 h in buffered 2.5% glutaraldehyde containing sucrose and post-fixated in buffered 2% OsO4 as previously described [22]. For the pre-embedding labeling of VLGR1 in adult zebrafish retinas and porcine retinas, the antibody against the C-terminal of murine VLGR1 (see Section 2.2. Antibodies) was used in our previously published protocol of pre-embedding labeling for TEM [23]. In brief, perforated fish eyes and eye cups of porcine eyes were pre-fixed in buffered 4% paraformaldehyde, dissected, infiltrated with 30% buffered sucrose, and cracked by several freezing-thawing cycles. After embedding in buffered 2% Agar (Sigma-Aldrich) agar blocks were sliced with a Leica VT1000S vibratome. Endogenous peroxidase activities of vibratome sections were suppressed by incubation with H_2_O_2_. Vibratome sections were incubated with primary antibodies against VLGR1 for 96 h and overnight with anti-rabbit biotinylated secondary antibody, which were visualized by a Vectastain ABC-Kit (Vector Laboratories, Newark, CA, USA). Retina sections were then postfixed sequentially in buffered 2.5% glutaraldehyde and 0.5% OsO_4_. Dehydrated specimens were flatmounted between ACLAR^®^-films (Ted Pella Inc., Redding, CA, USA) in Renlam^®^ M-1 resin (Sigma-Aldrich). After heat-polymerization, flatmounts were and clued on top of empty Araldit blocks for ultrathin sectioning. Specimens were ultrthin-sectioned with an Ultracut S ultramicrotome (Leica) and collected on Formvar-coated copper or nickel grids. Sections were counter stained with heavy metals before analyzed and imaged in a Tecna12 BioTwin TEM (FEI) equipped with a SIS Mega-View3 CCD camera (Surface Imaging Systems) was used. Images were acquired with a charge-coupled camera (SIS Megaview3; Olympus Soft Imaging Solutions GmbH, Muenster, Germany), archived by analSIS (Olympus Soft Imaging Solutions GmbH), and processed with Adobe Photoshop CS® (Adobe Systems, San Jose, CA, USA).

### 2.13. Proximity Ligation Assay (PLA)

In situ proximity ligation assays (PLA) were performed using the Duolink^®^ In Situ Red Starter Kit Mouse/Rabbit (Sigma-Aldrich), according to the manufacturer’s protocol. Primary antibodies were incubated overnight at 4 °C. Anti-rabbit PLUS and anti-mouse MINUS secondary PLA probes were applied. The two complementary oligonucleotides were then hybridized, ligated and amplified by rolling circle amplification. Targeted proteins with a proximity under 40 nm result in a fluorescence signal.

### 2.14. Ca^2+^ Imaging

For imaging the Ca^2+^ signals in the ER and mitochondria, pCMV G-CEPIA1er and pCMV CEPIA2mt were expressed in the cells, respectively [17]. Images were acquired with an eclipse Ti microscope (Nikon, Tokyo, Japan) equipped with a 63xApo TIRF oil objective (1.49 NA; Nikon). Dyes were excited with a 488 nm laser (Coherent; MPB communications Inc., Pointe-Claire, QC, Canada). The exposure time was 25 ms. Images were captured by a sCMOS camera (Orca-Flash 4.0 Hamamatsu Photonics, Herrsching, Germany) controlled by NIS-Elements Advanced Research acquisition software (Nikon). Experiments were performed at 35 °C. Extracellular solution contained, in mM: NaCl, 145; KCl, 2.5; glucose, 24; HEPES, 10; MgCl2, 2, CaCl2, 2; the pH was 7.3–7.4. Ca^2+^ transients were evoked by the addition of 10 μM ATP. Data were analyzed using Fiji, Prism (GraphPad, San Diego, CA, USA) and Microsoft Excel. The fluorescence values were normalized to the first 10 s of the measurement F/F0 = F(t)/F(t0 − 10s).

## 3. Results

### 3.1. Affinity Proteomics Identified Interactions of VLGR1 with Proteins of MAMs

To identify novel interaction proteins of VLGR1, we applied tandem affinity purification (TAP) [18,24]. We performed standard TAPs in highly transfectable HEK293T kidney cells and confirmed them in hTERT-RPE1 cells derived from the retinal pigment epithelium of the human eye. Both cell types were transfected with VLGR1_CTF constructs tagged with Strep II/FLAG (SF) (Figure 1B). Since tags can interfere with the interaction of molecules especially in the region of the tags and we aimed to identify proteins interacting with the N- and the C-terminal end of the VLGR1 molecule, we used N- and C- terminal tagged VLGR1_CTFs for our TAPs. After SF-TAPs eluted protein complexes were separated by liquid chromatography coupled with tandem mass spectrometry (LC-MS/MS) to determine their peptide contents. The raw spectra were searched against SwissProt databases to identify interacting proteins and results were verified in the Scaffold program. Next, proteomic data sets were analyzed by Software Tool for Researching Annotations of Proteins (STRAP) [25] and prey proteins were grouped according to their Gene Ontology (GO) terms by using a Cytoscape (http://www.cytoscape.org/) plugin (ClueGO; accessed on 10 September 2017) and STRING data (https://string-db.org/) (STRING; accessed on 20 March 2022) [26]. Our analysis revealed GO terms´ enrichments of proteins related to the endoplasmic reticulum (ER) and mitochondria for both cell types used (Figure 1C–E) (Appendix A). 257 proteins for VLGR1_CTF baits in HEK293T cells and 94 in hTERT-RPE1 cells, with an overlap of 81 proteins were identified. As mitochondria-related proteins we found 157 preys in HEK293T and 69 in hTERT-RPE1 cells, all of them were also present in the HEK293T pool (Figure 1C,D). Interestingly, we identified 30 TAP preys in HEK293T and 19 in hTERT-RPE1 cells, which associate to MAMs (Figure 1E) (Table 1). Among them are core MAM molecules related to the structural arrangement and formation of MAMs (5 proteins), such as the reticulon 4 protein (RTNA4, neurite outgrowth inhibitor or Nogo-A) and the vesicle-associated membrane protein-associated protein B (VAPB), which both are known to participate in ER-mitochondria tethering [21,27]. Other MAM-related prey proteins are known to be engaged in important MAM functions, namely the regulation of lipid metabolism (8 proteins) and/or Ca^2+^ homeostasis (9 proteins) or are implicated in tethering the membranes of the ER and mitochondria at MAMs (Table 1). Examples of identified molecules associated with lipid metabolism are long-chain fatty acid CoA ligase 4 (ACSL4), SLC27A4, or the two phosphatidylserine synthases PTDSS1 and PTDSS2. VLGR1 TAP preys associated with Ca^2+^ homeostasis are the sigma-1 receptor (S1R), which regulates the inositol trisphosphate receptor (IP3R) that mediates Ca^2+^ release from the ER upon stimulation. The two voltage-gated anion channels VADC1 and VADC2 in the outer mitochondrial membrane are responsible for the Ca^2+^ uptake of the mitochondria [28]. Cytoscape analysis demonstrated that the prey proteins identified in the VLGR1 TAPs related to the ER, mitochondria, and MAM cluster in specific protein networks (Figure 1E).

### 3.2. VLGR1 Interacts with MAM-Associated Proteins

Next, we validated the interaction of VLGR1_CTF with the MAM core proteins, namely ACSL4 and the S1R, previously described as ER residents [12,53], and the mitochondrial import receptor subunit TOM20, a component of the mitochondrial outer membrane [54]. In RFP-Trap^®^ immunoprecipitations of mRFP-tagged VLGR1_CTF expressed in HEK293T cells endogenous S1R, ACSL4 and TOM20 were recovered confirming their interaction with VLGR1 (Figure 2A,C,D). However, S1R was not recovered by mRFP-tagged VLGR1_ICD, indicating an interaction of the transmembrane or loop domains of VLGR1 with S1R (Appendix A). None of the MAM-associated proteins was co-precipitated with mRFP. Reciprocal Myc-Trap^®^ immunoprecipitation of Myc-tagged S1R and mRFP-tagged VLGR1_CTF co-expressed in HEK293T cells, confirmed the interaction of S1R and VLGR1_CTF (Figure 2B).

Next, we validated the interaction between overexpressed VLGR1_CTF and endogenous VLGR1 with S1R in HEK293T cells by in situ proximity ligation assays (PLA) (Figure 2E–G). We observed positive PLA signals and close proximity of VLGR1 and S1R in both scenarios, confirming the interaction of the two proteins in cells. No signals were observed in either control PLA. The interaction of VLGR1 and S1R observed in HEK293T cells was confirmed in Hela cells (Appendix A). Taken together, these findings strongly support the molecular interaction of VLGR1 with MAM core proteins.

### 3.3. VLGR1 Is Enriched in ER and MAM Fractions

We next wanted to verify the presence of VLGR1 in the MAM compartment by applying a cell fractionation assay modified from Wieckowski et al. [19] (Appendix A and Figure 2H). HA-tagged VLGR1-CTF transfected HEK293T cells were fractionated and the obtained fractions were validated by Western blots for specific marker proteins [19,55]. The MAM marker S1R was detected in the enriched MAM fraction [12]. As expected, S1R was additionally found in the whole cell lysate and in the fraction of crude mitochondria, which was also positive for the mitochondria marker TOM20 and in the ER fraction specified by the presence of the ER-resident protein CLIMP63 [56]. VLGR1_CTF was detected together with the MAM core proteins S1R and ACSL4 in crude mitochondria fraction, the ER and more importantly also in the fraction enriched for MAMs. This finding supports that VLGR1 is a component of the ER and the MAM.

### 3.4. VLGR1 Is Localized at the ER-Mitochondria Interface

Next, we aimed to validate the presence of VLGR1 in the MAM compartment by immunocytochemistry. For this, we immunostained the ER-resident protein Nogo-A/RTN4 and the outer mitochondrial membrane protein TOM20 which were both present in the VLGR1_CTF TAPs as prey, in HeLa cells expressing VLGR1_CTF_HA (Figure 3A). Immunofluorescence microscopy showed the co-staining of VLGR1 with the ER marker Nogo-A, confirmed by a positive Pearson correlation coefficient value (R = 0.70) and an overlapping staining with the outer mitochondrial membrane marker TOM20 with a positive but lower Pearson correlation coefficient (R = 0.33). Next, we determined the localization of the ER-mitochondria interface in HeLa cells by applying the “subtract function” tool in Adobe Photoshop^®^ subtracting the immunofluorescences signal of TOM20 and Nogo-A (Figure 3A). Merging the immunofluorescence of VLGR1 with the fluorescence subtraction result revealed the co-localization of VLGR1 with the ER-mitochondrial interface, which was also confirmed by the Pearson correlation coefficient (R = 0.40). The higher magnification of the zoom-ins of the merged images (Figure 3A, bottom panel) and the common peaks in the fluorescence intensity plots of the three channels of the region of interest (Figure 3B) further support these results. These findings indicate that VLGR1 is localized in ER membranes associated with ER-mitochondria interface of the MAM compartment illustrated in Figure 3C. The localization of VLGR1 at the interface of MAMs was confirmed by immunoelectron microscopy in sections through the inner segments of retinal photoreceptor cells of two different vertebrate species, namely the domestic pig (Sus scrofa) and zebrafish (Danio rerio) (Figure 3D).

### 3.5. VLGR1 Deficiency Alters the Structure of the ER-Mitochondria Interface

Characteristic changes in the MAM structure have been described in several diseases whose pathophysiology is associated with a MAM perturbation [11,57,58]. To determine whether VLGR1 deficiency also affects MAM structure, we focused on cerebral neurons and photoreceptor cells, both neurons whose dysfunction has been linked to neurologic disorders, such as epilepsy [59] as well as USH [11,57], respectively. To evaluate whether the deficiency of VLGR1 also affects the MAM structure, we examined MAMs in neurons of wild-type (WT) and *Vlgr1*delta7TM mice lacking the Vlgr1_CTF by transmission electron microscopy (Figure 4 and Appendix A). It has been previously highlighted that the contact point distances at the ER-mitochondrial interface are a key structural parameter related to MAM functions [57]. We measured the latter parameter in ultrathin sections through neurons of the cerebellum and the inner segment of retinal photoreceptor cells (Figure 4A and Appendix A). Quantitative morphometric analyses revealed that the contact point distances at the ER-mitochondrial interface were significantly enlarged in both cerebellar neurons and photoreceptor cells of *Vlgr1*del7TM mice when compared to wild-type controls.

Furthermore, we morphometrically analyzed two additional structural parameters of MAMs, namely the mitochondrial surface coverage by the ER and the mitochondrial perimeter in both cerebellar neurons and photoreceptor cells (Figure 4B,C and Appendix A). While the mitochondrial surface coverage by the ER was increased in both types of neurons, the mitochondrial perimeter was only increased in cerebellar neurons and not in photoreceptor cells of *Vlgr1*del7TM mice when compared to wild-type controls.

### 3.6. Ca^2+^ Transients at ER-Mitochondria Contact Sites Are Impaired by VLGR1 Deficiency

The MAMs represent a critical platform for the Ca^2+^ transfer from ER to mitochondria [60,61]. To monitor the Ca^2+^ flux in the two organelles, we made use of genetically encoded Ca^2+^ indicators targeted to ER, (G-CEPIA1er) and mitochondria (CEPIA2mt) [17]. We previously showed that primary astrocytes from mouse brain are well suited for studies of VLGR1 in cells [3,9]. Both Ca^2+^ indicators were transfected into primary brain astrocytes, derived from *Vlgr1*del7TM or wild-type control mice. To trigger the release of Ca^2+^ from the ER and to promote the mitochondrial Ca^2+^ uptake at the ER-mitochondria interface, we applied adenosine triphosphate (ATP) to the culture medium of the astrocytes. ATP stimulates purinergic receptor P2Y receptors, which activate phospholipase C (PLC), leading to hydrolysis of phosphatidylinositol 4,5-bisphosphate (PIP2) to diacylglycerol (DAG) and inositol triphosphate (IP3). Binding of IP3 to IP3 receptor (IP3R) on the ER membrane triggers the release of Ca^2+^ from the ER [62,63]. We monitored the dynamics of Ca^2+^ fluxes in the ER and the mitochondria in response to ATP in G-CEPIA1er- and CEPIA2mt-expressing astrocytes of *Vlgr1*del7TM and WT by live-cell imaging (Figure 5 and Appendix A). We quantified the time course of Ca^2+^ release from the ER (Figure 5A) by calculating the amplitude and the kinetics of the G-CEPIA1er signal, respectively (Figure 5B,C). The experiments showed significantly decreased Ca^2+^ release from the ER (Figure 5B), which was accompanied by an almost significant slowing of Ca^2+^ efflux in *Vlgr1*del7 TM astrocytes compared with wild-type control astrocytes (Figure 5C). Further, the measurement of CEPIA2mt signals showed a significant decrease of mitochondrial Ca^2+^ uptake in *Vlgr1*del7TM astrocytes (Figure 5D,E and Appendix A). In conclusion, our data demonstrate that VLGR1 is necessary for the proper regulation of Ca^2+^ homeostasis at MAMs and absence of VLGR1 results in dysregulated Ca^2+^ homeostasis at MAMs, as illustrated in Figure 5F.

## 4. Discussion

We identified the adhesion GPCR (aGPCR) VLGR1 in internal membranes of the cell, namely at the ER membrane and at mitochondria-associated ER membranes (MAMs), a specialized compartment at the interface between the membranes of the ER and mitochondria. MAMs provide platforms for several cellular processes and signaling pathways, such as ER stress signaling, metabolism, autophagy, apoptosis, inflammation, and Ca^2+^ homeostasis [10]. VLGR1 functions have been previously related to cell membrane adhesion, such as the ankle-links between the stereocilia of mechanosensitive hair cells in the inner ear and the periciliary membrane complex at the base of the cilium of photoreceptor cells or as a metabotropic mechanoreceptor at focal adhesions, the adhesion sites of the cell membrane to the substrate [9,14,64]. However, hitherto it has not been associated with functions at internal membrane contact sites of the cell, where opposing organelles are tethered facilitating the communication between organelles [65]. Here, we provide evidence that the aGPCR VLGR1 is a component of a protein complex that is crucial for regulating Ca^2+^ flux from the ER into the MAM interface to mitochondria and thus for the Ca^2+^ homeostasis.

We identified several MAM core proteins as interaction partners of VLRG1 by affinity proteomics capture approaches performed in two different human cell lines. All MAM protein hits that we discovered in hTERT-RPE1 were also present in the HEK293T dataset, providing evidence that VLGR1 is closely associated with MAMs. We confirmed the putative interactions with several MAM components of both the ER part and the outer mitochondrial membrane using complementary protein-protein interaction assays, such as in vitro co-IPs and in situ PLAs. In addition, our observed subcellular localization of VLGR1 in the ER membrane and its enrichment in the intersections between the ER and mitochondria further supports that VLGR1 is a component of MAMs. All in all, our data demonstrate the molecular association of VLGR1 with MAMs and its localization in the mitochondria-ER contacts of MAMs in the membrane of the ER.

MAMs play pivotal roles in cellular physiology by regulating intracellular Ca^2+^ homeostasis driven by a tightly balanced interplay between the Ca^2+^ release from the ER and the Ca^2+^ uptake through the juxtaposed outer mitochondrial membrane [60,61,66]. To this end, MAMs harbor a channel complex for Ca^2+^, mainly composed of the inositol 1,4,5-trisphosphate receptors (IP3Rs) in the ER membrane, bridged by the chaperone GPR-75 to the voltage-dependent anion channel 1 (VDAC1) in the outer mitochondrial membrane [67], which channels Ca^2+^ to the mitochondrial Ca^2+^ uniporter (MCU), and the Na^+^-dependent mitochondrial Ca^2+^ efflux transporter (NCLX), both of the inner mitochondrial membrane [28]. In the present study, we provide evidence that VLGR1 is closely associated with this Ca^2+^ delivery system at MAMs by interacting with several of its components. We demonstrate that VLGR1 physically interacts with the sigma-1 receptor (S1R), the Ca^2+^-sensitive and ligand-operated receptor chaperone which regulates the Ca^2+^ out-flux via IP3Rs from the ER at MAMs [39]. In addition, we show the interaction of VLGR1 with TOM20, a peripheral subunit of the mitochondrial outer membrane translocase essential of mitochondrial protein import. It has been recently shown that TOM20 forms a complex with the VDAC1 and cooperates in channeling Ca^2+^ ions into mitochondria [68]. It is worth noting that both voltage-dependent anion channels VDAC1 and VDAC2 were also found to be hits in the VLGR1 TAPs (see Table 1), so they also represent potential binding partners of VLGR1 in a protein complex related to VLGR1 in the MAM compartment (see cartoon, Figure 2E).

In addition to these physical molecular interactions of VLGR1, we observed a functional interaction of VLGR1 with the Ca^2+^ signaling between the ER and mitochondria at MAMs. In the absence of VLGR1, both the Ca^2+^ release from the ER through IP3Rs in response to ATP and the subsequent Ca^2+^ uptake by mitochondria were significantly reduced. We assume that the reduced outflow of Ca^2+^ from the ER into the cleft at the MAM interface does not provide a sufficiently large pool of local Ca^2+^ available for uptake by mitochondria. The result is that the absence of VLGR1 leads to reduced uptake of Ca^2+^ into the mitochondria. This is consistent with the altered mitochondrial Ca^2+^ dynamics observed after siRNA-mediated knockdown of S1R and IP3R3, respectively [39]. Even without direct measurements of the Ca^2+^ release from the ER, the authors of the latter study assumed that the measured reduced Ca^2+^ uptake by mitochondria was also caused by a disturbed Ca^2+^ supply from the neighboring ER Ca^2+^ store. Overall, our results suggest that VLGR1 participates in the regulation of IP3Rs in the ER membrane of MAMs, directly or through its interaction with the membrane chaperonin S1R which controls the functions of IP3Rs.

Under normal physiological conditions, the membranes between the ER and the mitochondrial membranes are approximately 10–30 nm apart at the MAM interfaces [69,70]. This close arrangement is required to ensure proper flow of Ca^2+^ and normal exchange of phospholipids between the two organelles. Several ER-mitochondria tethering protein complexes consisting of proteins located on the opposing membranes have been reported to establish and maintain this distance. Measuring the contact point distances at the ER-mitochondrial interface in cerebellar neurons and photoreceptor cells revealed a significant increase in the distances between the ER and mitochondria in the absence of VLGR1 in both neurons and photoreceptor cells (Figure 4A). This might be due to a direct association of VLGR1 with the tethering complexes, which is supported by the identification of ER and mitochondrial tethering molecules as interaction partners of VLGR1, namely VDAC1, the VAMP-associated protein (VAPB) or reticulon 4 (RTN4, Nogo-A) both are present in our VLGR1 TAPs (Table 1) [21,27,70]. The wider distance between the membranes of both organelles at MAMs in VLGR1 deficient cells could be, however, also due to reduced Ca^2+^ release from the ER and the resulting decrease of the local Ca^2+^ concentration in the cleft. Because of many molecular links, the assembly by the tethering molecules is Ca^2+^-dependent [71].

Several diseases have been associated with alterations in the ER-mitochondrial junctions as a phenotype. Alterations in the connections between the ER and the mitochondria have been described as a pathological feature underlying the pathomechanisms of many neurodegenerative diseases, such as Alzheimer’s disease, Parkinson’s disease, and amyotrophic lateral sclerosis with associated frontotemporal dementia [70]. Here, we show that the absence of VLGR1 leads to a comparable phenotype at MAMs. Thus, the pathophysiological pathways of neurodegenerative diseases and the two diseases associated with VLGR1 defects, Usher syndrome type 2 and childhood absence epilepsy [4,6,7], may open the avenue for common therapeutic targets and therapy options.

One obvious pathway that is disrupted in these diseases is the bioenergetics of cells. Due to the reduced Ca^2+^ influx, the Ca^2+^ level in the mitochondria decreases, which impairs the activity of mitochondrial enzymes, such as the pyruvate, isocitrate, and α-ketoglutarate dehydrogenases, and thereby the bioenergetics of the cells [72,73]. This is in line with previous findings indicating that the pathology in Alzheimer’s and retinal degeneration are accompanied by effects on the mitochondrial energetics [74,75].

## 5. Conclusions

We identified and validated VLGR1 as the first aGPCR in MAMs. We show that VLGR1 is a vital component for the correct architecture of MAMs. In addition, we demonstrate that VLGR1 is crucial for balancing the Ca^2+^ homeostasis of MAMs, mitochondria and thereby the entire cell. Our findings also provide novel insights that help to explain the pathomechanisms underlying VLGR1-associated diseases, namely the human Usher syndrome type 2C and childhood absence epilepsy.

## Figures and Tables

**Figure 1 cells-11-02790-f001:**
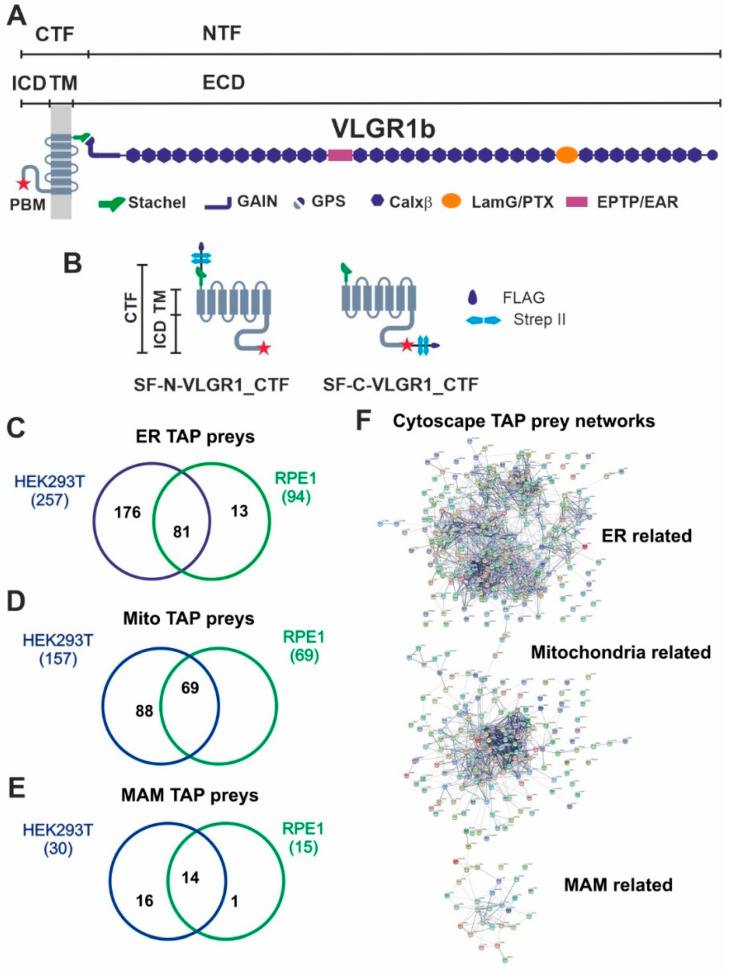
VLGR1 protein domain composition, VLGR1_CTF constructs for TAPs and overlap of proteins associated to ER, mitochondria and MAM identified in VLGR1_CTF TAPs. (**A**) VLGR1 protein domain composition: VLGR1 molecules consist of a C-terminal fragment (CTF) and a N-terminal fragment (NTF) which result from the cleavage at the highly conserved GPCR proteolytic site (GPS) in the GAIN (autoproteolysis-inducing) domain positioned next to the seven-span transmembrane domain (TM). After autoproteolytic cleavage the first 11 amino acids of VLGR1_CTF, the so-called Stachel peptide, can act as a tethered internal agonist for receptor activation [3]. The extracellular domain (ECD) of VLGR1b contains numerous Ca^2+^ binding calcium exchanger β motifs (Calx-β), seven epilepsy-associated/Epitemptin-like (EAR/EPTP) repeats and a pentaxin/laminin G-like domain (LamG/PTX). The intracellular domain (ICD) ends with a class-1 PDZ binding motif (PBM). (**B**) VLGR1_CTF, N- or C-terminal tagged with Strep II/FLAG (SF)-tag used as baits in TAPs. (**C**) Venn diagram shows an overlap of 81 proteins of ER-related TAP prey for polled VLGR1_CTF N-and C-terminal tagged TAPs from HEK293T and hTERT-RPE1 cells. (**D**) Mitochondria-related prey proteins from VLGR1_CTF N-and C-terminal tagged TAPs in HEK293T and hTERT-RPE1 cells, compared in a Venn diagram with an overlap of 69 proteins. (**E**) 14 common MAM prey proteins are identified in TAPs from HEK293T and hTERT-RPE1 cells. (**F**) Protein networks of ER, mitochondria and MAM-related proteins, illustrated by Cytoscape analysis of preys identified in VLGR1_CTF N-and C-terminal tagged TAPs.

**Figure 2 cells-11-02790-f002:**
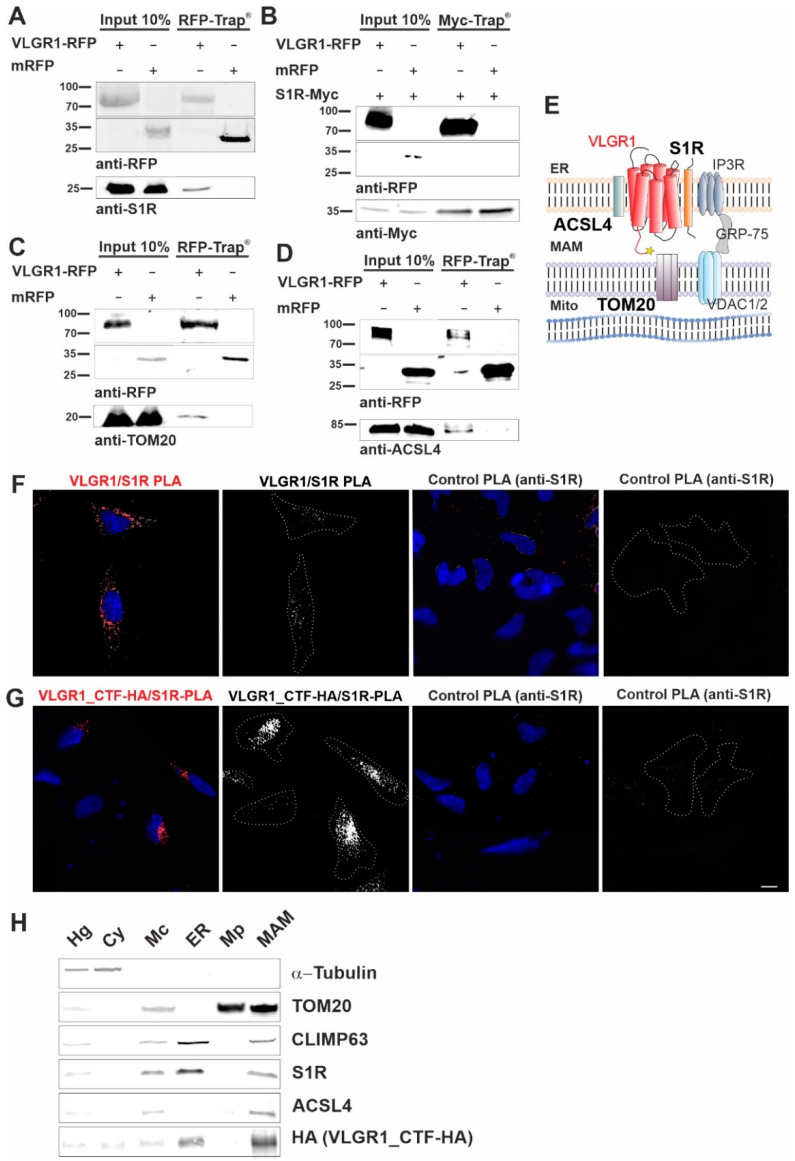
VLGR1_CTF interacts with S1R, ACSL4, TOM20 and is present in the purified MAM compartment. (**A**–**C**) Western blot analysis of RFP-Trap^®^ from lysates of HEK293T cells expressing VLGR1_CTF-mRFP or mRFP. Endogenous S1R (sigma-1 receptor), ACSL4 (long-chain fatty acid CoA ligase 4) and TOM20 are precipitated by VLGR1_CTF-mRFP. (**D**) Western blot of Myc-Trap^®^ from HEK293T cells coexpressing VLGR1_CTF-mRFP or mRFP and S1R-Myc. VLGR1_CTF-mRFP binds to S1R-Myc. (**E**) Schematic illustration of interactions complex of VLGR1 at the MAM site: pull-downs show that VLGR1 interacts with the ER membrane molecules ACSL4 and S1R. S1R binds to IP3R (inositol trisphosphate receptor), regulating the Ca^2+^ effluxes through the receptor. IP3R is in turn connected via GRP75 to voltage-gated anion channels VDAC1/2 in the outer mitochondrial membrane, where they are associated with the mitochondrial importer subunit TOM20 which also interact with VLGR1. Mito, mitochondria. (**F**) Proximity ligation assay (PLA) for endogenous VLGR1 and S1R in HEK293T cells. White PLA signals represent colocalization of both proteins. Both, anti-S1R and anti-VLGR1 (not shown) only were probed with both PLA (rabbit, rb and mouse, ms) secondary antibodies including PLUS and MINUS complementary sequences and used as negative control. (**G**) PLA assay for VLGR1_CTF-HA and S1R in VLGR1_CTF-HA transfected HEK293T cells. Protein-protein interaction is indicated by white PLA signals. Anti-S1R and anti-VLGR1 (not shown) with PLA (rabbit, rb and mouse, ms) secondary antibodies including PLUS and MINUS complementary sequences were used as negative control, no signal was observed in the control. All images of the respective replicates were acquired using the same light exposure and intensity settings. Nuclei were stained with DAPI (blue). Scale bar, 10 µm. (**H**) Representative Western blot analysis of fractions obtained in MAM isolation assay with VLGR1_CTF-HA transfected HEK293T cells. Purity of cellular fractions was validated by marker proteins, namely α-Tubulin for the cytoplasm, CLIMP63 for the ER. TOM20 as markers for the inner and outer mitochondrial membrane, respectively, and sigma-1 receptor (S1R) and ACSL4 as the MAM component [12]. VLGR1_CTF is present in isolated/crude mitochondria (Mc), ER and MAM fractions. Cell homogenate (Hg), crude mitochondria (Mc), endoplasmic reticulum (ER), pure mitochondrial fraction (Mp), mitochondria-associated membranes (MAM) and cytosolic (Cy). A total of ~30 µg of protein was loaded for each fraction. The pulldown experiments shown in (**A**–**C**) and the PLAs shown in (**F**,**G**) were each performed three times. The fractionations for MAM isolation analyzed in (**H**) were carried out five times.

**Figure 3 cells-11-02790-f003:**
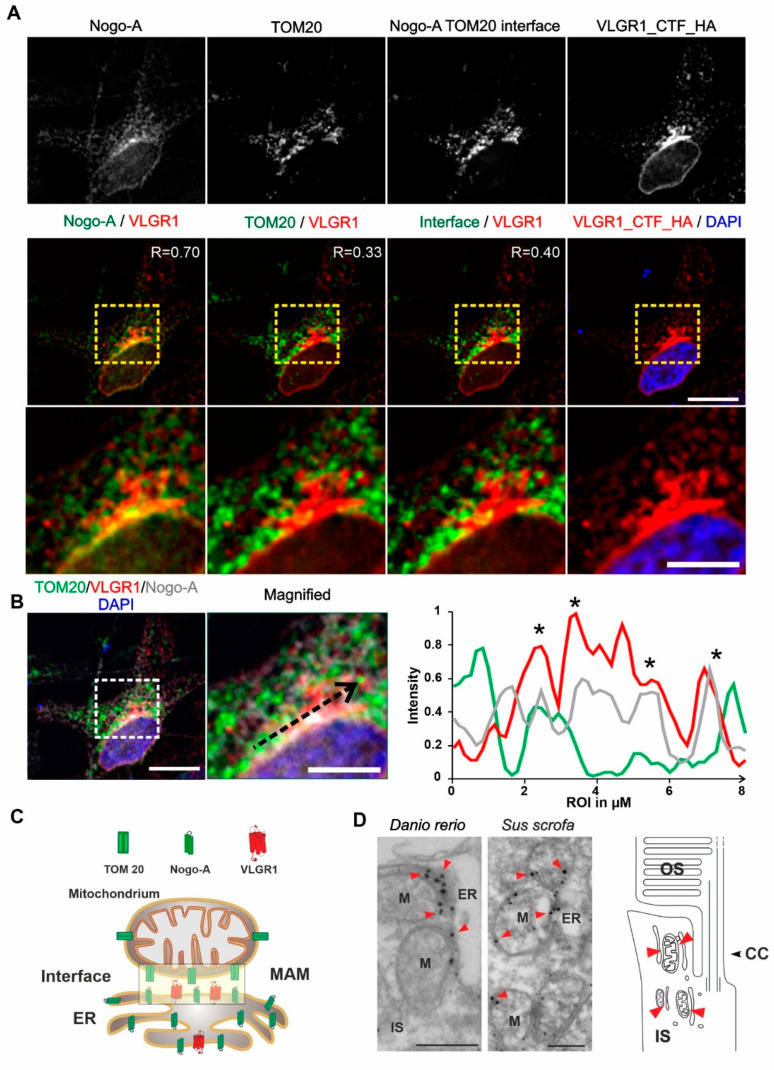
VLGR1_CFT localizes at the ER-mitochondria interface and is present at mitochondrial membranes in photoreceptor cells. (**A**) Immunocytochemical localization analysis of VLGR1_CTF at the ER-mitochondria interface in HeLa cells. Noga-A and TOM20 were used as markers for the ER and for the outer membrane of mitochondria, respectively. ER-mitochondria interface is visualized by the overlay of Nogo-A and TOM20 signals and subtraction of all individual Nogo-A and TOM20 signals. VLGR1_CTF-HA single staining and merged with DAPI (blue). VLGR1_CTF (red) is localized at the ER-mitochondria interface (Pearson coefficient 0.40). VLGR1_CTF is also localized at the ER with a Pearson coefficient of 0.70 and at mitochondria with a Pearson coefficient of 0.33. Pearson coefficients were calculated for 50 cells in three independents experiments. (**B**) Triple immunofluorescence of VLGR1_CTF-HA (red), TOM20 (green), and Nogo-A (grey), counterstained with DAPI (blue) for nuclear localization. Normalized intensity plot of the tree channels of region of interest (ROI) indicated by the black dashed arrow in the magnified image. Shared intensity peaks (asterisks) indicate the co-localization of VLGR1_CTF-HA with TOM20 and Nogo-A. (**C**) Schematic representation of Nogo-A and TOM20 distribution in ER and mitochondria and along the interface of the two organelles. VLGR1 localization is indicated at the ER-mitochondrial interface. (**D**) Immunoelectron microscopy of the inner segment (IS) of retinal photoreceptor cells from a zebrafish (*Danio rerio*) and a domestic pig (*Sus scrofa*) showing localization of VLGR1 along the membranes of mitochondria (M) (red arrowheads) in close proximity to the ER. Cartoon of a rod photoreceptor cell demonstrating the localization of mitochondria-ER interfaces (red arrowheads). Connecting cilium (CC) and the outer segment (OS) are indicated. Scale bars: (**A**) 10 µm, 5 µm; (**B**) 10 µm, 5 µm; (**C**) 400 nm.

**Figure 4 cells-11-02790-f004:**
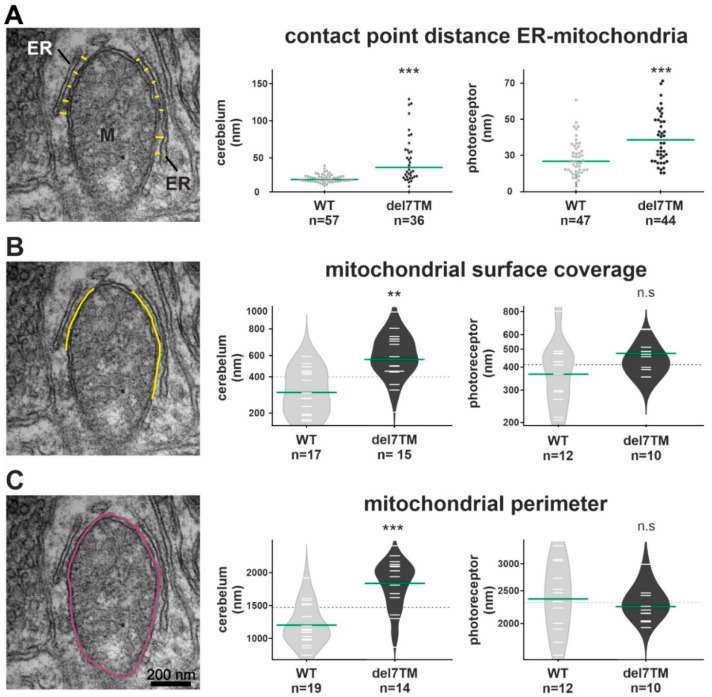
Electron microscopy analysis of the ER-mitochondria interface in murine cerebellum neurons and retinal photoreceptors and S1R abundance at the ER. (**A**) Representative TEM image for the measurement of contact point distance at the ER-mitochondrial interface (WT mice). Mitochondria (M) and the ER (yellow spacer lines) are indicated. Calculation of contact point distance at the ER-mitochondrial interface in WT control and *Vlgr1*del7TM cerebellar neurons and photoreceptor cells. (**B**) The mitochondrial surface coverage by ER is indicated by yellow lines in the representative TEM image. Measurements of mitochondrial surface coverage in WT control and *Vlgr1*del7TM cerebellum neurons and photoreceptor cells. (**C**) The mitochondrial perimeter indicated in purple in the representative TEM image was measured in WT control and *Vlgr1*del7TM cerebellum neurons and photoreceptors. Three independent samples of mature mice of both genotypes and sexes at the age of 4 to 6 months were analyzed. Student’s t-test for statistical significance: ** *p* < 0.01, *** *p* < 0.001, n.s. no significance.

**Figure 5 cells-11-02790-f005:**
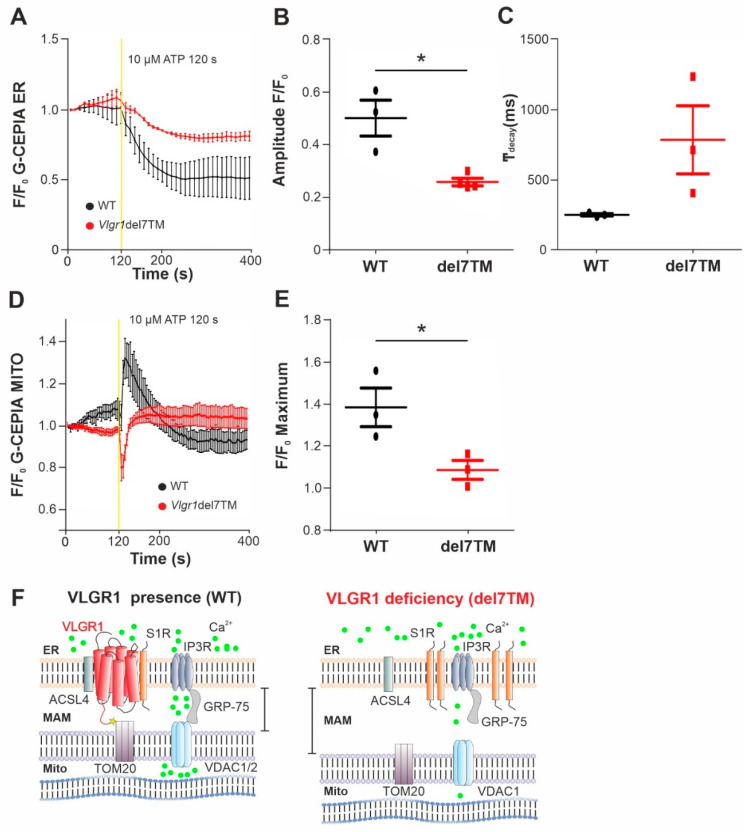
VLGR1 deficiency decreases the Ca^2+^ release from ER and uptake to mitochondria in astrocytes. (**A**) Averaged time-course of the ER Ca^2+^ signal in response to ATP stimulation at 120 s, monitored by G-CEPIA1er, in WT control (black) and *Vlgr1*del7TM *(red)* astrocytes. (**B**) Quantification of G-CEPIA1er signal amplitude, *p* = 0.0056. WT: *n* = 3 individual experiments; *Vlgr1*del7TM: *n* = 3 individual experiments. (**C**) Quantification of time constant (τ) of the G-CEPIA1er signal decay (fitting by single exponential function), *p* = 0.2814. (**D**) Mitochondrial Ca^2+^ signals were measured using CEPIA2mt after stimulation with ATP at 120 s in WT control and *Vlgr1*del7TM astrocytes. (**E**) Amplitude of signals for mitochondrial Ca^2+^ response after ATP stimulus; *p* = 0.0438. WT: *n* = 3 individual experiments; *Vlgr1*del7TM: *n* = 4 individual experiments. Yellow line indicates application of 10 µM ATP at 120 s. Error bars represent SEM. Student’s t-test for statistical significance: * *p* < 0.05 (**F**) Schematic illustration of Ca^2+^ release (green dots) from the ER through and mitochondrial uptake of Ca^2+^ by the Ca^2+^ shuttling complex of MAMs in the presence and absence/deficiency of VLGR1. VLGR1 interacting partners identified TAPs and confirmed in pull-downs: ACSL4, long-chain fatty acid CoA ligase 4; S1R, Sigma-1 receptor; TOM2, mitochondrial import receptor. IP3R (inositol trisphosphate receptor) is regulated by S1R and bridged by GRP75 to the voltage-dependent anion channels VDAC1 and 2, both also present as prey in VLGR1_CTF TAPs.

**Table 1 cells-11-02790-t001:** MAM related proteins identified by VLGR1 TAPs in HEK293T and hTERT-RPE1 cells proteins associated with: Ca^2+^ homeostasis, orange; lipid metabolism, yellow, MAM structure, gray.

Gene	Protein	Protein Function	Reference	HEK	RPE1
** *ACSL4* **	Long-chain-fatty-acid—CoA ligase 4	lipid biosynthesis & fatty acid degradation	[29]	+	−
** *AIFM1* **	Mitochondrial apoptosis-inducing factor 1	apoptosis, mito. morphology	[30]	+	−
** *AMFR* **	Autocrine motility factor receptor	ubiquitination	[31]	+	−
** *BCAP31* **	B-cell receptor-associated protein 31	apoptosis	[32]	+	−
** *BSG* **	Basigin	regulatory component of γ-secretase	[33]	+	+
** *CANX* **	Calnexin	calcium pump	[34]	+	+
** *CISD2* **	CDGSH iron sulfur domain 2	calcium homeostasis	[35]	+	−
** *ERLIN2* **	Erlin-2	targets IP3Rs for degradation	[36]	+	+
** *ERP44* **	ER resident protein 44	ER protein retention	[37]	+	−
** *G6PC3* **	Glucose-6-phosphatase 3	gluconeogenesis	[38]	+	−
** *HSPA5* **	Heat shock protein 5	chaperone, ER stress	[39]	+	+
** *HSPA9* **	Mitochondrial heat shock protein 9	chaperone, binds VDAC	[40]	+	+
** *LCLAT1* **	Lysocardiolipin acyltransferase 1	chardolipin acyl chain remodeling	[41]	+	−
** *LMAN1* **	ERGIC-53	mannose-specific lectin	[42]	+	−
** *MAVS* **	Mito. antiviral-signaling protein	activation of NF-kb/IRF3	[43]	+	+
** *P4HB* **	Protein disulfide-isomerase	ER protein retention	[44]	+	+
** *PIGN* **	Phosphatidylinositol glycan anchor biosynthesis, class N	glycolipid, GPI syntheses	[41]	+	−
** *PSEN1* **	Presenilin-1	component of γ-secretase	[45]	+	−
** *PSEN2* **	Presenilin-2	component of γ-secretase	[45]	+	−
** *PTDSS1* **	Phosphatidylserine synthase 1	phospholipid metabolism	[46]	+	−
** *PTDSS2* **	Phosphatidylserine synthase 1	phospholipid metabolism	[46]	+	−
** *RHOT1* **	Mitochondrial Rho GTPase 1	mitochondrial trafficking	[47]	+	−
** *RTN2* **	Reticulon-2	formation of tubular ER	[48]	+	+
** *RTN4* **	Reticulon 4, neurite outgrowth inhibitor, Nogo-A	ER-mitochondria tethering, membrane trafficking	[21]	+	+
** *SCD* **	Stearoyl-CoA desaturase	fatty acid transport	[49]	+	+
** *SIGMAR1* **	Sigma-1 receptor, SR1	chaperone, lipid transport, Ca^2+^ signaling	[39]	+	+
** *SLC27A4* **	Long-chain fatty acid transport protein 4	fatty acid transport	[50]	+	+
** *SOAT1* **	Sterol O-acyltransferase 1	cholesterol metabolism	[42]	+	+
** *VAPB* **	VAMP-associated protein	ER-mitochondria tethering	[27]	+	−
** *VDAC1* **	Voltage-dependent anion-channel 1	ion exchange, Ca^2+^ transport	[40]	+	+
** *VDAC2* **	Voltage-dependent anion-channel 2	ion exchange, Ca^2+^ transport	[51]	−	+
** *WSF1* **	Wolframin, ER, transmembrane	Ca^2+^ transport, ER stress	[52]	+	−

## Data Availability

All data obtained in the present work are included in the main body of the publication or are provided as Appendix A.

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
