# Peer review of "The Adhesion GPCR VLGR1/ADGRV1 Regulates the Ca2+ Homeostasis at Mitochondria-Associated ER Membranes"

_cells, 2022, doi:10.3390/cells11182790_

Round 1

Reviewer 1 Report

See PDF file

Reviewer 2 Report

Reviewer’s comments:

The manuscript titled “The Adhesion GPCR VLGR1/ADGRV1 regulates the Ca2+ homeostasis at mitochondria-associated ER membranes“ authored by Krzysko et al describes the use of a proteomic approach to identify the physical interaction of the Adhesion GPCR VLRG1/ADGRV1 with components of the ER and mitochondria membranes as well as components of the mitochondria-associated membranes. The authors confirm some of VLGR1’s key interactions using biochemical approaches (pull-downs assays) and through proximity labelling (Proximity ligation assays) but also identify the receptor’s endogenous subcellular localization to compartments located between mitochondria and ER in tissues. The subcellular localization-dependent functionality of VLGR1’s is further evidenced using cells from genetically engineered VLGR1 7TM-deficient mice (cerebellar neurons, photoreceptor cells, astrocytes) by evaluating the receptor’s loss-of-function influence on ER and mitochondrial calcium transients as well as mitochondrial morphology and interface area with the ER. While the study provides a yet unknown and novel role for GPCRs in general and VLGR1 in particular, the manuscript seems incomplete as various sections or results are either not described or simply absent. We recommend that the authors address the following comments before the manuscript can be resubmitted in its revised version:

Minor revisions:

-       Figure 4D is not described nor referred to in the main text. Please complete the results section by adding this description.

-       While siRNA-mediated VLGR1 knockdown procedures are described in the Methods, the authors do not describe the experiment in which it was used nor the results obtained. Please provide the corresponding information in the results section.

-       Figure 5 panels B and C are incorrectly described in the figure legends (interchanged). Additionally, the tau parameter should be analyzed for the experiments done in Figure 5D.

-       As per Cells’ instructions to authors, please provide the full-length images of western blotting analysis and include them as a single supporting information file.

-       Please briefly explain in the main text the rationale for using both N- and C-terminal SF-tagged VLGR1 constructs concurrently in TAP experiments

-       Please indicate parameters and details for reproducibility purposes:

o   number of independent experiments [Figure 2, Figure 3, Figure 4]

o   definition of “n” in each figure (number of cells, number of independent experiments, etc)

o   Technical replicates where relevant

-       Methodological details:

a.    Please detail the centrifugation steps conducted for isolation of MAM fractions

b.    Please describe types of cerebellar neurons that were selected and analyzed for TEM experiments

c.     Please specify the age of animals used in the study (gender also if relevant)

d.    Please describe in the methods the F/F0 equation and the optical parameters underlying the acquisition of calcium transients using the calcium indicators (excitation wavelength, exposure time, emission wavelength, etc)

-       Transmission electron microscopy: Please indicate antibody reactivity region for VLGR1 detection

-       Figure 2H, the term “Mp” is not described in the figure legend. Please provide the missing information.

-       Figure 5A is not referred to in the main text. Please include the corresponding reference.
